# Disinhibition Is an Essential Network Motif Coordinated by GABA Levels and GABA B Receptors

**DOI:** 10.3390/ijms25021340

**Published:** 2024-01-22

**Authors:** Nelson Villalobos

**Affiliations:** 1Academia de Fisiología, Escuela Superior de Medicina, Instituto Politécnico Nacional, Plan de San Luis y Díaz Mirón, Colonia Casco de Santo Tomás, Ciudad de México 11340, Mexico; nvillalobosv@ipn.mx; 2Sección de Estudios Posgrado e Investigación de la Escuela Superior de Medicina, Instituto Politécnico Nacional, Plan de San Luis y Díaz Mirón, Colonia Casco de Santo Tomás, Ciudad de Mexico 11340, Mexico

**Keywords:** circuit, neurophysiology, oscillation, GABA B receptor, GABA levels, disinhibition, network, tonic inhibition, network dynamics, gain

## Abstract

Network dynamics are crucial for action and sensation. Changes in synaptic physiology lead to the reorganization of local microcircuits. Consequently, the functional state of the network impacts the output signal depending on the firing patterns of its units. Networks exhibit steady states in which neurons show various activities, producing many networks with diverse properties. Transitions between network states determine the output signal generated and its functional results. The temporal dynamics of excitation/inhibition allow a shift between states in an operational network. Therefore, a process capable of modulating the dynamics of excitation/inhibition may be functionally important. This process is known as disinhibition. In this review, we describe the effect of GABA levels and GABA_B_ receptors on tonic inhibition, which causes changes (due to disinhibition) in network dynamics, leading to synchronous functional oscillations.

## 1. Introduction

The neural network is the substrate of brain function. The functional dynamics of neural networks modify the synaptic physiology, leading to the reorganization of these circuits [1]. These modifications modulate the firing of local units that determine the network’s functional state [2]. In the network’s functional state, neurons respond to both convergent and divergent inputs (from the network itself, recurrent local networks, or even distal networks), which can be excitatory or inhibitory. These dynamics effectively influence the output signals by modulating selective connections, leading to constant changes in the network pathways [3]. Thus, the balance between excitation and inhibition (E/I) is a fundamental parameter that influences the network’s output signal [4]. An imbalance in this parameter impacts the progression of various diseases [5,6].

Networks have a stable state in which they are not entirely inactive. In this state, the neuronal activity (including synaptic strength, intrinsic properties, and E/I balance) produces a broad spectrum of networks with various properties (size, dynamics, and linkage frameworks [7]). In an operational network, the temporal evolution between E/I is proportional among individual neurons and across the global network. This feature allows rapid transitions between states that translate into functional output signals [3]. Therefore, a process able to modulate the network state by controlling the temporal dynamics between E/I is functionally essential. This process is known as disinhibition. Defined as a selective and transient hindrance to inhibition that leads to excitation [8], disinhibition is an evolutionarily conserved process involved in various motor [9,10] and cognitive functional processes, brain regions, neuronal types, and projection neuron compartments [8,11]. A crucial physiological property is its duration, which can range from milliseconds to days [8,12].

Another essential signal modulation mechanism is gain control, which involves regulating the amplitude caused by the E/I balance [13]. In this framework, inhibition in the perisomatic region controls the gain of projection neuron responses, which modulates the arrangement of synaptic inputs [8]. In the network, gain control induces normalization of the average neuron firing rate according to the inputs [8,14,15]; therefore, inhibition adjusts the gain. Consequently, a regenerative depolarization that begins in the dendrites is the process by which distal excitatory inputs lead to neural activation; this process has been described in relation to disinhibition [16].

Neural network oscillations are involved in multiple physiological processes and are accepted as tools for communication among brain regions. Thus, oscillations lead to the organization and coordination of information due to the precise interactions among the activities of different neurons over time. The emergence of oscillations and their frequency ranges depend on both the intrinsic neuronal properties and the network properties [2,17].

Gamma-aminobutyric acid (GABA) is a canonical inhibitory neurotransmitter. GABA performs its function through three types of receptors: GABA_A_, GABA_B_, and GABA_C_. The versatility of GABA is demonstrated not only in the diversity of receptors but also in how it establishes inhibition, with phasic and tonic forms. In the phasic form, the postsynaptic GABA_A_ receptor is activated by an increase in GABA in the synaptic cleft after release. In the tonic form, GABA overflows from the synaptic cleft and activates extrasynaptic GABA A receptors (located in the presynaptic terminal and synapses with adjacent neurons), allowing temporally and spatially slow transmission. The extrasynaptic receptors show a high affinity to GABA and mainly constitute the *δ* subunits assembled at *α*4 or *α*6 [18,19,20,21]. The importance of extrasynaptic GABA_A_ receptors (GABA_A_ Rs) to the tonic current is widely accepted [20] and evidence of the involvement of GABA_B_ receptors (GABA_B_ Rs) as modulators of this current has been presented [22].

According to the above information, in the present review, we describe results that suggest that both GABA levels and GABA_B_ Rs influence disinhibition, which modulates the gain mode of the output signal and thus changes the dynamics of the network (observed as changes in oscillations), thus allowing a functional state.

## 2. Involvement of GABA B Receptors

GABA_B_ Rs are ubiquitous metabotropic receptors in the brain. These receptors couple with the Gi protein. Consequently, the G_*α*i_/G_*α*o_ subunits inhibit adenylyl cyclase (AC), which reduces cAMP levels and thus decreases protein kinase A (PKA) activity [21]. The G_*βγ*_ subunits inhibit the Ca^2+^ channels and activate the GIRK and TREK-type K^+^ channels [23]. The affinity of the G_*βγ*_ subunit to GIRKs increases due to the Na^+^ that enters during the action potential; this increase in affinity leads to the opening of the channel [24].

GABA_B_ Rs are expressed at both inhibitory and excitatory synapses. They require two different subunits for their function: GABA_B1_ and GABA_B2_. GABA binds to the GABA_B1_ subunit and the GABA_B2_ subunit causes signaling [25]. Due to gene splicing, GABA_B1_ has two isoforms, GABA_B1a_ and GABA_B1b_, which have different N-N-terminal regions. This region determines that the GABA B1a isoform is expressed at the presynaptic terminal in excitatory synapses, thus modulating glutamate release [25,26,27]. However, both isoforms are expressed at the postsynaptic terminal. The GABA_B1b_ subunit in this terminal allows coupling with the K^+^ channels, reducing this ion’s current [28]. Moreover, this subunit inhibits dendritic Ca^2+^ spikes by affecting voltage Ca^2+^ channel (VGCC) sensitivity [28]. This molecular diversity leads to heterogeneous responses that dynamically modulate synaptic transmission and, therefore, the network.

### 2.1. Presynaptic Modulation

The critical presynaptic function of GABA_B_ Rs is inhibiting the release of neurotransmitters by restricting the entry of Ca^2+^ into the terminal, followed by the inhibition of VGCC by the G_*βγ*_ subunit. However, this is not the only mechanism of GABA_B_ Rs. GABA_B_ Rs also modulate the release mechanism at various levels. For example, they modulate the SNAP-25 protein by decreasing cAMP levels, which reduces vesicular priming. During this process, the G_*βγ*_ subunit modulates neurotransmitter release through direct interaction with the SNAP-25 protein, and this interaction is modulated by synaptotagmin-1 [29]. Furthermore, neurexins have been reported to enable GABA_B_ Rs to function at this terminal [30]. Therefore, the activation of presynaptic GABA_B_ Rs by reducing GABA release leads to the formation of a negative feedback loop, which is typical of autoreceptor-mediated synaptic gain control [31].

GABA_B_ Rs exert gain modulation at the network level through their presynaptic control. In olfactory inputs, they control differential presynaptic gain [32]. In the presynaptic neurons of the inferior colliculus, GABA_B_ Rs control the excitability gain, and blockading them increases adaptation to the stimulus [33]. In the auditory pathway, they normalize the activity of the neuronal population by controlling the gain [34]. Presynaptic GABA_B_ Rs change the E/I ratio of the neurons in the prefrontal cortex, thereby modulating network excitation in a model of autism [35].

### 2.2. Postsynaptic Modulation

The predominant effect at the postsynaptic level is the inhibition mediated by the activation of the GIRK channels by the G_*βγ*_ subunit [36]. Once active, the G_*βγ*_ subunit forms a complex with the RGS protein, which binds to the GIRKs and accelerates their kinetics [37]. At the same time, the RGS protein increases the GTPase activity of the G*α* subunit, causing rapid desensitization of the K^+^ current [38], which reduces neuronal excitability and inhibits action potential backpropagation. Furthermore, GABA_B_ Rs activate a two-pore domain K^+^ channel (TREK2, also called KCNK10 [39,40]). In the dendrites, the joint activation of GIRK, TREK2, and VGCC and the inhibition of action potential backpropagation [41] reduce the generation of Ca^2+^ spikes (Figure 1A).

### 2.3. Modulation of Network Dynamics

GABA_B_ Rs modulate the dynamics in the network by decreasing the output current. Once activated, GABA_B_ Rs inhibit the N-type calcium and BK-type potassium channels. Consequently, the neurons increase their degree of depolarization, incrementing their excitation level [23]. It has been suggested that inhibiting the N-type channels leads to operational advantages by expanding the transmission dynamics without influencing neurotransmitter release [23]. In the lateral geniculate nucleus, the activation of GABA_B_ Rs leads to strong hyperpolarization, followed by rebound firing as mediated by the T-type Ca^2+^ channels, which improves the signal detectability while altering sensory discrimination [42]. GABA_B_ R activation in the network is related to burst firing and constant rhythmic activity. In this way, GABA_B_ Rs may temporally modulate slow network activity, the strength of fast activity, and the relative firing during network oscillations [43].

## 3. GABA Levels

Recent evidence suggests that the synaptic levels of GABA are essential to synaptic transmission and network physiology. Quantitative reports suggest that the synaptic GABA levels are in the mM range, while the extrasynaptic levels are in the μM range. However, in the nuclei of specific neurons, these levels are higher at rest and during physiological processes [36]. Consequently, it is currently accepted that wiring-based (synaptic) and volume-based (non-synaptic) transmissions both occur in the brain [44,45]. This context supports the proposition that the molecular versatility of GABA has functional implications for the network physiology due to its effect on different neuronal compartments. In this way, GABA levels modulate the output signals and considerably influence the network dynamics.

Accepted functional roles of the GABA levels in both synaptic transmission and the network dynamics include the generation of tonic current and the activation of extrasynaptic GABA_A_ and GABA_B_ Rs (including subtypes of both receptors sensitive to low GABA concentrations [46]). This current has been found in multiple synaptic compartments, including the presynaptic terminals (inhibitory and excitatory), the perisynaptic regions of the excitatory synapses, and the middle of unmyelinated axons and dendrites [45].

It is evident that GABA (by controlling the generation of membrane potential with different forms in various regions) modulates the temporal integration of synaptic inputs and, consequently, the activity patterns of the neuronal populations that make up the circuits, with the tonic current playing a fundamental role. Thus, the stimulating current is modulated by different processes, including the sustained activation of individual cells, the coordination of presynaptic events of different interneurons, an increase in the current density at the release site, and the reuptake mechanisms, which depend on the transporters.

GABA transporters (GATs) are widely expressed in multiple nuclei, compartments, and neuronal glia, with a higher expression in axons than dendrites [47]. The function of GATs depends on several factors. First, the amplitude of the receptor activation is transiently modulated by the GABA concentration. Second, GATs control the receptor kinetics during the recruitment of neighboring synapses. Third, GATs are involved in spatial transmission; they convert spatially confined inhibitory signals into waves without spatial restrictions, which can activate type A or B receptors at both the presynaptic and postsynaptic terminals [46,48]. Thus, GATs functionally participate in the modulation of tonic currents. They reduce the conductance of neuronal inputs, temporarily modulating the incoming excitatory outputs. Therefore, tonic GABA currents control the input–output relationship (neuronal stimulation versus response) through shunt inhibition [49] along the dendrite or axon.

The contribution of GABA Rs to tonic current has been described. Studies on neurons in the locus ceruleus suggest that GABA_B_ Rs generate tonic currents through ERK1-dependent activation [50]. In the basal amygdala, an increase in tonic inhibition related to presynaptic regulation was reported [51]. Interestingly, increases in the tonic current due to the postsynaptic activation of GABA_B_ Rs have been reported in thalamocortical neuron granule cells in both the cerebellum and the dentate gyrus. In the dentate gyrus, the increase in tonic current was related to an increase in membrane traffic and the expression of high-affinity extrasynaptic GABA_A_ Rs induced by GABA_B_ R activation (Figure 1A) [51].

Hefty evidence in the last decades shows the physiological role of glial cells in synaptic transmission, especially astrocytes [52,53]. Astrocytes transform glutamatergic excitation into GABAergic inhibition [54] and contribute to establishing negative feedback [55,56]. During synaptic transmission, astrocytes absorb glutamate, which releases GABA. The GABA released establishes tonic inhibition depending on the network activity and modulates the E/I balance [57].

## 4. Disinhibition

Functionally, a microcircuit may have a “disinhibiting” motif, including serial connections between two inhibitory interneurons and a principal excitatory neuron (Figure 1B) [58]. These circuits are involved in the response to social fear. In this process, the increase in the activity of somatostatin (SST)-positive interneurons (INs) inhibits parvalbumin-positive (PV+) INs, which causes disinhibition of the principal cells of the dorsomedial prefrontal cortex [59]. Similarly, an exaggerated response to fear is mediated by a disinhibitory circuit involving the dorsal raphe nucleus, pericentral reticulotegmental nucleus, and central reticulotegmental nucleus [60].

An interesting disinhibitory process was described in the lateral entorhinal cortex. In it, optogenetic silencing of VIP INs (positive for vasoactive intestinal peptide) significantly decreased the incidence of the dendritic spikes driven by the lateral entorhinal cortex, suggesting a disinhibitory effect being exerted on the dendritic activity by the INs [61]. A disinhibitory process at the dendritic level was described in the hippocampus associated with the LTP process [62]. The VIP INs in the hippocampus regulate LTP through disinhibition by activating the VPAC1 receptors (G protein-coupled receptors of the VIP/PACAP family). This molecular mechanism affects the expression and phosphorylation of the K_v4.2_ K^+^ channels in the dendrites of the hippocampal pyramidal cells [11].

During attentional selection, selective disinhibition improved the target firing rates with resemblance to multiplicative input gain, another commonly reported effect of attention on neural responses [63]. In the cerebellum, the activation of GABA_B_ Rs in the dentate gyrus (DG) improves the granule cell (GC) activity by reducing the synaptic inhibition imposed by hilar INs. This disinhibitory action facilitates the transfer of signals from the hippocampus. Furthermore, the GABA_B(1a,2)_ and GABA_B(1b,2)_ R subtypes differentially modulate the GC output through dendritic locations and axon terminals. The disinhibition described during spatial and pattern learning in the hippocampus is mediated exclusively by GABA_B(1a,2)_ Rs. A disinhibitory circuit of the distal dendrites of pyramidal cells mediates motor integration in the somatosensory cortex [64]. Furthermore, it was recently reported that both GAT-1 blockade and GABA_B_ agonism disinhibited the neurons in the reticular thalamic nucleus (RTn) [65,66].

Several research results support the functional importance of inhibition in neuronal circuits [67,68]. Therefore, it is accepted that inhibition stabilizes the circuit, thus allowing calculations to be carried out that include network stability, response normalization, and input amplification through a process called inhibitory plasticity [67,68,69]. This process modulates both the generation and firing frequency properties of the excitatory neurons and the input–output function. In the circuit, excitation is propagated by rapid changes in input, the downregulation of inhibitory gains, or temporary changes in the E/I balance. In this way, it establishes coordinated communication between circuits. Thus, disinhibition by increasing the firing frequency allows the transfer of information between circuits. This mechanism has been proposed as the basis of association memory. Through E/I modulation, disinhibition allows the recovery of memories, which are kept in a resting state [68]. Studies that analyze the involvement of the GABA levels, as well as the participation of GABA_B_ Rs, are on the rise.

In addition to various physiological processes, disinhibition has been proposed as a pathophysiological substrate of various diseases. In Parkinson’s disease, disinhibition in the primary motor cortex has been suggested as one of the first manifestations [70] and is correlated with freezing and gait impairment [71]. Obesity is associated with disinhibition of the orbitofrontal cortex. It has been shown that excitability is secondary to the reduction in the tonic inhibition of the pyramidal neurons and the excitability decreases with increasing GABA levels [72].

Tourette’s syndrome (TS) is a hyperkinetic disorder characterized by motor and phonetic tics [73,74]. However, its clinical presentation is commonly accompanied by various abnormal behaviors (motor, sensory, and complex behavioral), such as inappropriate non-obscene behaviors, impulsivity, obsessive-compulsive disorder, and attention deficit hyperactivity disorder [74] The central pathophysiological substrate is disinhibition [10,73,74]. This originates from alterations in the GABAergic signaling and a loss of E/I balance in the cortex–basal ganglia–thalamus–cortex circuit associated with the modulation of motor output, motor learning, and action selection. In TS, alterations in the E/I balance leads to a reduction in the gain in motor cortical excitability. In this context, increasing GABA levels in the somatosensory cortex alter the gain of excitability (by increasing tonic inhibition), which improves the control of motor outputs [73].

The disinhibitory circuit is ubiquitous: its presence is not limited to the cerebral cortex and it communicates with subcortical structures. It has an important implication in regulating muscle tone, which allows motor control [75]. Regarding GABA_B_, it is suggested that these receptors are extrasynaptic and are tonically activated by the GABA released by astrocytes as well as by the GABAergic interneurons in the spinal motoneurons and descending afferent fibers [76]. A transcendental role of disinhibition has been described in the pathophysiology of pain, especially chronic pain [77,78]. In its development, a loss of inhibition as mediated by parvolabumin-positive interneurons has been suggested, as well as simultaneously the postsynaptic inhibition of the vertical cells. These events allow the pain circuits of lamina I to be activated by innocuous tactile stimuli [77].

## 5. Oscillation

Various compelling studies have described the modulatory participation of GABA_B_ Rs in slow cortical oscillations. During slow wave oscillations (SWOs), GABA_B_ R blockade modifies three important aspects of the SWO cycle. First, GABA_B_ R blockade increases the number of “up” states; second, GABA_B_ R blockade affects the subsequent duration of the “down” state; and third, GABA_B_ R activation desynchronizes the SWOs [79]. The participation of GABA_B_ Rs according to their synaptic location has also been described. At the presynaptic level, GABA_B_ Rs contribute to spontaneous transitions from the down state, while postsynaptic receptors are essential for the afferent termination of the up state. Thus, GABA_B_ Rs containing the GABA subunit contribute to spontaneous termination of the up state, and GABA_B_ Rs containing the GABA_B1b_ subunit are essential for afferent evoked termination of the upstate [80]. Furthermore, GABA_B(1a,2)_ heteroreceptors selectively located at the thalamocortical relay (TCR)–RTn cell synapses in the thalamus (Th) regulate the oscillation strength, while GABA_B(1b,2)_ Rs control the oscillation frequency [81].

Thalamic oscillations of 3 to 5 Hz are characteristic of absence epilepsy. In a rodent model, selectively blocking the GABA transporters ([GAT] to GAT-1 or GAT-3) was shown to prolong the oscillations; however, blocking both transporters inhibited the oscillations. In this same model, it was reported that extending the activity in a narrow range of GABA_B_ Rs promoted the opening of T-type channels and intensified the oscillations [82].

In addition, tonic inhibition and disinhibition have been suggested to regulate motor activity [83]. An analysis of the cortical oscillations after GAT-1 blockade demonstrated that high GABA levels influence the beta oscillations related to movement [84]. Moreover, it was recently reported that high GABA levels and the participation of GABA_B_ Rs in the external globus pallidus disinhibit the RTn and thus desynchronize the beta oscillations in the motor cortex [65,66]. In the human visual cortex, the GABA levels modulate the gamma oscillations. Increased GABA levels in this region (after GAT-I blockade) lead to decreased gamma oscillations [85]. In the hippocampus, activation of the GABA_B_ Rs selectively suppresses the recruitment of SST Ins during gamma oscillations induced in vitro (Figure 1B) [86].

## 6. Discussion

The neurophysiological aspects addressed in this paper form an essential part of the physiology of the neuronal dynamics. Thus, the data described firmly support the participation of both the GABA levels and GABA_B_ Rs in neuronal dynamics.

The recent results regarding cortical disinhibition are undeniable [87]. However, the current data regarding cortical disinhibition focus on cortical vision, leaving aside the subcortical structures. Thus, it is widely accepted that the subcortical structures part of the interneuronal network are exclusively inhibitory. In this organization, the data described suggest that convergent afferent synaptic activity can alter the precise temporal arrangement of the network activity through disinhibition. Furthermore, the flow of such information toward a functional neuronal network is highly regulated by synaptic inhibition as mediated by the GABA_B_ Rs [88], contributing to the disinhibition process.

Thus, to establish communication between circuits, the phase of the incoming message must be the disinhibitory phase of the oscillation of the receiving circuit [3]. Therefore, in network dynamics, disinhibition could be a mechanism that opens the door to sending information. The current results support the proposition that the exit gate leads to the dendrites, as described in the entorhinal cortex (EC). This region contains a long-range disinhibitory motif that facilitates the integration of inputs from the EC into the hippocampus, increasing the probability of dendritic spiking while inducing the input’s time-dependent plasticity [89].

In network dynamics, the dendrites process information from multiple inputs; simultaneously, they can segregate them into various regions [90]. To generate dendritic spikes, various events must occur, such as intense and repetitive activity, integration with other coincident inputs, and inhibition blockage. Thus, it has been proposed that a disinhibitory circuit participates as a gain modulation mechanism that allows dendritic nonlinearities in a rapid, dynamic, and path-specific manner. Furthermore, it increases the dendritic excitation further when combined with long-range disinhibitory stimuli. During oscillatory synchrony between the EC and the hippocampus, high-amplitude dendritic spikes that do not produce action potentials in the soma are observed [90]. An excitatory process that integrates the GABA_A_ and _B_ receptors was proposed several years ago [91]. It was suggested that in thin dendritic processes, massive activation of the GABA_A_ Rs would lead to depolarizing values. At the same time, the K^+^ inversion potential would be altered, which would produce inefficiency in the activity of the GABA_B_ Rs. This inactivity would indicate that the equilibrium potential of K^+^ moves toward the resting potential, increasing the PEPS amplitude without causing hyperpolarization. Consequently, GABA_B_-type inhibition in large distal dendrites silences the GABA_A_-mediated inhibition [91].

In microcircuits, information processing between inhibitory internal neurons that modulate inhibitory and excitatory operations is carried out according to lateral inhibition processes. One such process involves the RTn–Th–cortical (TC) circuit. In this circuit, the axonal fields of the RTn neurons and the dendritic domains of the TCs overlap. Based on this morphological characteristic, it has been suggested that disinhibitory feedback loops form due to disinhibition, increasing the contrast between the central and peripheral neurons generated by lateral inhibition [92]. Thus, both processes (lateral inhibition and disinhibition) can establish feedback loops, leading to communication between circuits.

In conclusion, GABA Rs and GABA levels both contribute to the formation of exit gates related to network communication in the cortex and subcortical regions, and these processes modulate motor and cognitive events at various levels.

## Figures and Tables

**Figure 1 ijms-25-01340-f001:**
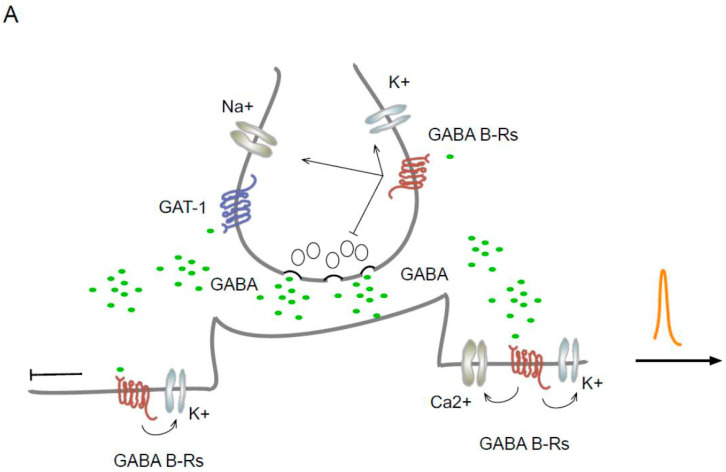
Localization of GAT-1 and GABA_B_ Rs in synaptic regions and a disinhibitory circuit. (**A**) Schematic representation of the localization and effect of GAT-1 and GABA_B_ Rs. In dendrites, the GABA_B_ receptor activates the GIRK, TREK2, and VGCC-type channels and inhibits the backpropagation of the action potential while reducing the generation of Ca^2+^ spikes. (**B**) The scheme represents a disinhibitory circuit as a mechanism for activating specific pathways and its effect on oscillation. The microcircuit includes serial connections between two inhibitory interneurons and a main excitatory neuron and their effect on oscillation. The thick lines represent disinhibition.

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
