# Peer review of "Disinhibition Is an Essential Network Motif Coordinated by GABA Levels and GABA B Receptors"

_ijms, 2024, doi:10.3390/ijms25021340_

Round 1
Reviewer 1 Report
Comments and Suggestions for Authors
This is a very basic view of the area the author reviews, not too deep and perhaps lacking novel insight into how studies in the field can best advance.
My major comment is the total non-compliance with the traditional nomenclature used to designate GABAA and GABAB receptors. I highly recommend following the standard nomenclature for those receptors. A search for symbols in the literature will show what the standard stance is for them. The same applies to the nomenclature of ions and genes. I strongly recommend the author to pay special attention to these details. Without them there seems to be a lack of understanding and clarity of the topic at hand. I strongly recommend reviewing the entire manuscript and correcting all omitted and erroneous nomenclature, in text and figures.
Comments on the Quality of English LanguageAcceptable
Reviewer 2 Report
Comments and Suggestions for Authors
This is a nice concise review focusing on examples of disinhibition in neural circuits. This will be beneficial to other researchers to have the literature compiled on the topic. Every few years new reviews are needed to bring the readership up to date. The text is easy to read. Some places the text is a bit choppy, but it is hard to compile so much literature into a story line without just presenting statements to highlight the main take home points. Overall, I feel this is a fine review.
1. Abstract: “Therefore, a process capable of modulating the temporal dynamics of excitation/inhibition and, consequently, the network state may be functionally important. This process is known as disinhibition.”
It is understandable that the authors is trying to write a concise abstract but the definition here does not seem to capture the real definition as defined in the Introduction.
2. Line 136: 3. Gaba levels. Capitalize GABA
3. Therea are some other reviews on Disinhibition that the authors might want to investigate as some of the concepts highlighted are similar. It might also be interesting to mention the concept that disinhibition maybe a state in the Tourette Syndrome.
Wu YK, Miehl C, Gjorgjieva J. Regulation of circuit organization and function through inhibitory synaptic plasticity. Trends Neurosci. 2022 Dec;45(12):884-898. doi: 10.1016/j.tins.2022.10.006. Epub 2022 Oct 28. PMID: 36404455.
Kurvits L, Martino D, Ganos C. Clinical Features That Evoke the Concept of Disinhibition in Tourette Syndrome. Front Psychiatry. 2020 Feb 25;11:21. doi: 10.3389/fpsyt.2020.00021. PMID: 32161555; PMCID: PMC7053490.
4. Other studies do address disinhibition not just in the cortical regions but also in the spinal cord for controlling motor communication. Maybe some mention of other locations outside the brain would be of interest to a reader.
Hughes DI, Todd AJ. Central Nervous System Targets: Inhibitory Interneurons in the Spinal Cord. Neurotherapeutics. 2020 Jul;17(3):874-885. doi: 10.1007/s13311-020-00936-0. Epub 2020 Oct 7. PMID: 33029722; PMCID: PMC7641291.
Round 2
Reviewer 1 Report
Comments and Suggestions for Authors
No further comments.